# A Meaningful Strategy for Glioma Diagnosis via Independent Determination of hsa_circ_0004214

**DOI:** 10.3390/brainsci13020193

**Published:** 2023-01-23

**Authors:** Yinan Zhou, Yating Zhang, Jiajia Tian, Zengli Miao, Shangrui Lv, Xudong Zhao

**Affiliations:** 1Department of Neurosurgery, Medical School of Nantong University, Nantong University, Nantong 226019, China; 2Wuxi No. 2 People’s Hospital, Affiliated Wuxi Clinical College of Nantong University, Wuxi 214002, China; 3Wuxi Clinical Medical College of Nanjing Medical University, Nanjing Medical University, Wuxi 214002, China; 4Department of Neurosurgery, Wuxi No. 2 People’s Hospital, Wuxi 214002, China

**Keywords:** circular RNA, hsa_circ_0004214, glioma, diagnostic markers, molecular network

## Abstract

Glioma is one of the most common primary tumors in the central nervous system. Circular RNAs (circRNAs) may serve as novel biomarkers of various cancers. The purpose of this study is to reveal the diagnostic value of hsa_circ_0004214 for glioma and to predict its molecular interaction network. The expression of hsa_circ_0004214 was evaluated by RT-qPCR. The vector and siRNAs changed the expression of hsa_circ_0004214 to judge its influence on the migration degree of glioma cells. hsa_circ_0004214 can be stably expressed at a high level in high-grade glioma tissue (WHO III/IV). The area under the ROC curve of hsa_circ_0000745 in glioma tissue was 0.88, suggesting good diagnostic value. While used to distinguish high-grade glioma, AUC value can be increased to 0.931. The multi-factor correlation analysis found that the expression of hsa_circ_0004214 was correlated with GFAP (+) and Ki67 (+) in immunohistochemistry. In addition, the migration capacity of U87 was enhanced by overexpression of hsa_circ_0004214. Through miRNA microarray analysis and database screening, we finally identified 4 miRNAs and 9 RBPs that were most likely to interact with hsa_circ_0004214 and regulate the biological functions of glioma. Hsa_circ_0004 214 plays an important role in glioma, its expression level is a promising diagnostic marker for this malignancy.

## 1. Introduction

Glioma, the most common primary cancer in the central nervous system, is an aggressive and highly lethal disease [1], accounting for more than 80% of all primary brain tumors [2]. Glioma originates from the glial cells surrounding the central nerve and has been characterized by necrosis, aggressive growth, and angiogenesis. Even if patients with malignant glioma undergo common treatments, such as surgical resection, radiotherapy, and chemotherapy, the median survival time is only approximately 1 year [3]. Although some molecular pathological findings, such as isocitrate dehydrogenase 1 (IDH1) mutation and chromosome 1p/19q co-deletion status, are known to be predictors of prognosis, the molecular mechanism underlying occurrence and development remains unclear [4]. The clinical symptoms of glioma lack specificity, while common intracranial hypertension and neurological dysfunction usually occur later. The diagnosis of glioma often relies on imaging evidence, and the tumor type can be judged based on the location, shape, size, boundary, and nature of the tumor. The prognosis level and mortality of glioma are closely related to the WHO grade of glioma. However, there is a lack of general consensus on the molecular markers for glioma diagnosis and WHO grading, and glioma markers have changed greatly [5]. Therefore, it is very significant to definite diagnostic markers of glioma, which will implement the accurate diagnosis of glioma and find new therapeutic targets.

Endogenous noncoding RNAs include microRNAs (miRNAs), long noncoding RNAs (lncRNAs), and circular RNAs (circRNAs) [6]. Although there is plenty of evidence that noncoding RNAs play critical regulatory roles in various physiological and pathological processes, the molecular regulatory mechanisms remain obscure in terms of cancer progression and metastasis [7]. Recent findings suggest that circRNAs in tissues provide a new and potential tool for cancer diagnosis. Circular RNAs have been thought to be accidental by-products resulting from transcription errors [8]. Due to the widespread application of new technologies, circRNAs have been recognized and taken seriously in various biological fields. Numerous studies have shown that circRNAs are characterized by differential expression in many diseases [9]. According to this feature, intraoperative detection of differentially expressed circRNAs may be a potential supplement to the immunohistochemical diagnosis of glioma [10].

Recently, this study found an interesting circRNA, hsa_circ_0004214, which independently testing and assessing their expression in tissues will more accurately determine the presence of gliomas and the WHO classification of gliomas. Moreover, in this study, a novel circular RNA (hsa_circ_0004214) was found to be important for migration of tumors. By expanding the number of human glioma and control samples used in this study, we were able to analyze the expression of circRNAs that were independently detected in tissue samples from glioma patients. We analyzed the diagnostic value of hsa_circ_0004214 in different grades of glioma and evaluated the correlation between various pathological factors and hsa_circ_0004214. In addition, we preliminarily verified the value of hsa_circ_0004214 in judging the migration capacity of glioma cells in vitro, and constructed the molecular network that hsa_circ_0004214 regulates the occurrence and development of tumors.

## 2. Materials and Methods

### 2.1. Human Tissue Specimens and Ethical Approval

A total of 30 pairs of glioma tissue samples were collected, including 20 pairs of glioma samples matched with paracancerous tissue. In addition, 10 pairs of glioma samples matched with control cortex (non-glioma tissue as control group) were collected. All specimens came from surgery. In order to accurately collect glioma and paracancerous tissues, experienced doctors separated the tissues according to the intraoperative situation or navigation guidance. We tried to collect the glioma core tissue after excluding necrotic tissue, and the paracancerous tissue should be at least 1 cm away from the glioma on a safe basis. Non-glioma control brain tissue (10 cases) is usually collected with non-functional regions next to lesions of non-glioma patients. Fresh tissues were taken from the Neurosurgery Department of the Second People’s Hospital of Wuxi from 2019 to 2022. After the tissues were taken, the blood on the surface of the tissues was washed with normal saline, and stored in liquid nitrogen for more than 10 min. These actions were completed within 15 min at most. Finally, the samples were quickly transferred to a −80 °C freezer for long-term storage. The patient’s diagnosis was independently re-reviewed by 2 pathologists and classified according to WHO criteria. Written informed consent was obtained from all patients for this research. Tumor volumes were measured using preoperative MRI scans that were acquired on the day of or prior to surgery. All of these samples were obtained at the initial diagnosis. Each patient’s age, gender and clinical stage were recorded. The study was approved by the Ethics Committee of Wuxi Second People’s Hospital [Y-166].

### 2.2. Cell Culture and Transfection

Glioma cell line (U87) was purchased from the Stem Cell Bank, the Chinese Academy of Sciences. U87 cells were cultivated with DMEM high glucose (biosharp, Shanghai, China) supplemented with 10% fetal bovine serum (Gibco, Australia). U87 were incubated at 37 °C in 5% CO_2_. Use Lipofectamine 3000 (Invitrogen, Carlsbad, CA, USA) to transfect cells with designated nucleotides or plasmids, according to the manufacturer’s instructions. We constructed the overexpression vector of hsa_circ_0004214. We also constructed siRNA1 and siRNA2 of hsa_circ_0004214. The transfection process used Opti-MEM (1X, gibco, Shanghai, China). Plasmid and siRNA details are in the Appendix A. After 48 h of transfection, the next step is performed when the cells are in good condition.

### 2.3. Quantitative Real-Time Polymerase Chain Reaction (RT-qPCR)

Total RNA was isolated from tissues by TRIzol reagent (Invitrogen, USA) according to the kit instructions. The cDNA was synthesized with random primers using the Hifair II 1st strand cDNA synthesis kit (YEASEN Biotech, Shanghai, China). RT-qPCR uses Bio-Rad CFX96TM real-time PCR system, hieff qPCR SYBR Green Master Mix (YEASEN Biotech, Shanghai, China). Primers were synthesized by general biology (Anhui, China). Data were normalized to GAPDH. The relative expression of RNA was analyzed by 2-ΔΔCt or log2 method.

### 2.4. Fluorescence In Situ Hybridization (FISH)

The location of hsa_circ_0004214 in U87 was detected by FISH assay using fluorescence probes. Reagents included, 4% paraformaldehyde (Sigma, MKCL5723), 1% Triton x-100 (Biofroxx, 1139ML500), wet box, pre-hybridization solution (BOSTER, AR0152), oligonucleotide probe diluent (BOSTER, AR0062), DAPI staining solution (Beyotime C1005), anti-fluorescence quenching mounting medium (Beyotime, P0126). The Fish probe sequence of hsa_circ_0004214 is (5′ to 3′): GTTCTTGGCGTGCTGACTGG. The probe concentration was diluted to 500 nM, and denatured at 85 °C for 5 min. See the Appendix A for details. The Nikon AISi Laser Scanning Confocal Microscope (Nikon instru-ments Inc., Tokyo, Japan) was used to visualize the images.

### 2.5. Transwell Migration Assay

Transwell migration assay Transwell (Costar, Corning, NY, USA) with a multipolar (8.0 μm) polycarbonate membrane was utilized to conducted cell migration experiments. 5 × 10^4^ cells were mixed with serum-free medium and added into the upper chamber of the insert. Then, 800 μL complete medium was added to the bottom chamber. The U87 in the chamber were mixed in 6% paraformaldehyde for 10 min and stained with crystal violet. After 30 min, we removed the chamber and gently wiped the dye on the surface of the upper chamber. U87 in the lower chamber were preserved and the number of cells on the bottom surface was observed and analyzed under a microscope.

### 2.6. Bioinformatics Analysis

The biological analysis of hsa_circ_0004214 comes from circbase (http://www.circbase.org) (accessed on 10 April 2022). Databases of circBANK (http://www.circbank.cn) (accessed on 14 May 2022), CSCD (https://gb.whu.edu.cn/CSCD/) (accessed on 11 May 2022), ENCORI (https://starbase.sysu.edu.cn/) (accessed on 17 June 2022) and miRDB (http://mirdb.org/) (accessed on 11 May 2022) were crossed to forecast miRNA and RBP. The profiles of miRNA for human samples derived from patients with glioma were obtained from the GEO database (https://www.ncbi.nlm.nih.gov/geo/) (accessed on 28 May 2022). The GSE13030 miRNA microarray was selected based on preset criteria: (1) The expression profiles of miRNA in human samples were derived from glioma patients and matched normal tissues; (2) Candidate microarrays enrolled at least three pairs of samples. The downstream target gene interaction relationship of miRNAs comes from PubMed (https://pubmed.ncbi.nlm.nih.gov/) (accessed on 21 August 2022) and NCDB (https://www.facs.org/quality-programs/cancer-programs/national-cancer-database/) (accessed on 12 August 2022).

### 2.7. Statistical Analysis

All experimental data were analyzed using prism8.0 (GraphPad) and image J software. The expression level of every circRNA was represented as the 2-ΔΔCt method or log2 transformation. Differences of circRNA levels between glioma and peritumoral tissues, were calculated by *t*-test. The correlation between circRNA levels and clinicopathological factors was further analyzed by one-way analysis of variance. Data are presented as mean ± s.e.m. *p* < 0.05 was used to designate significant differences. F-test to assess the correlation between hsa_circ_0004214 and width of edema band around glioma. The receiver operating characteristic (ROC) curve was established to evaluate the diagnostic value. The cut-off value of hsa_circ_0004214 was analyzed with SPSSUA.

## 3. Results

### 3.1. hsa_circ_0004214 Stable Expression in Glioma

Characterization of hsa_circ_0004214 in glioma according to the human reference genome (GRCh37/hg19) acquired, we found that the genomic length of the hsa_circ_0003258 is 922 bp and the spliced length is 922 bp from the circBase database and NCBI genome database. hsa_circ_0004214 was formed by the back-splicing of exon 3 of linear gene angiomotin like 1 (AMOTL1). AMOTL1 is a member of the Motin protein family (Table 1). Meanwhile, AMOTL1 is associated with angiopoietin, an angiostatin-binding protein that regulates endothelial cell migration and capillary formation. PCR assay indicated that hsa_circ_0004214 could be amplified by outward-facing divergent primers in cDNA (Table 2). These data demonstrated that the hsa_circ_0004214 contained one circularized exons, which formed by exon back-splicing (Figure 1). FISH analysis revealed that the majority of hsa_circ_0004214 preferentially localized in the cytoplasm (Figure 2A). These results showed that hsa_circ_0004214 is a stable circRNA expressed in glioma.

### 3.2. High hsa_circ_0004214 Expression in Human Glioma Tissue

According to previous work, we have identified several circRNAs that were potentially differentially expressed in glioma core tissue. In this article, we focused on the expression level and diagnostic potential of hsa_circ_0004214 in glioma. Here, we investigated the expression of hsa_circ_0004214 in 20 pairs of glioma core tissues and corresponding paracancerous normal brain tissues by RT-qPCR. Considering the clear relationship between the experimental group and the paracancerous group, we defined ∆∆CP < −0.5 as high expression when counting the CP value of RT-qPCR. We found that the expression level of hsa_circ_0004214 in glioma core tissue was steadily increased compared to that in tumor peripheral tissue of the same glioma patient (Figure 2B, *p* < 0.05). After that, we divided 20 independent tissue samples from different patients into two groups, including a non-glioma control group (*n* = 10) and a glioma group (*n* = 10), and detected the expression of hsa_circ_0004214 in each tissue, respectively. The results showed that the expression level of the hsa_circ_0004214 was higher in the glioma group than in the non-glioma control group (Figure 2C, *p* < 0.01). The sensitivity and specificity of hsa_circ_0004214 differential expression in gliomas and paracancerous tissues in the same patient, were significantly steadier than those between non-glioma control group and glioma group collected from different patients (Table 3). The expression level of hsa_circ_0004214 was not significantly different between paracancerous tissues and non-glioma tissues (Figure 2D, *p* > 0.05). Relatively, there was no difference in the expression of hsa_circ_0004214 in central glioma tissues between the two groups (Figure 2E, *p* > 0.05). These results indicated that hsa_circ_0004214 was highly expressed in glioma.

### 3.3. Expression of hsa_circ_0004214 in Different Grades of Gliomas

To investigate whether hsa_circ_0004214 expression varied in different WHO grades, we detected hsa_circ_0004214 expression in 30 gliomas and 10 non-glioma controls. The glioma specimens were verified and classified according to the WHO grading standard of glioma in 2021 by two experienced clinical pathologists. The samples were divided into four groups, including the non-glioma control group (*n* = 10), WHO II (*n* = 5), WHO III (*n* = 10), and WHO IV (*n* = 15). The expression levels of hsa_circ_0004214 were analyzed and compared, respectively, among each group. Normality testing studies whether quantitative data (non-glioma control group, *n* = 10) analysis had normal distribution properties. Shapiro-Wilk test is recommended for small samples (n < 30). Evaluating specifically at the expression level of hsa_circ_0004214 (*p* = 0.398), indicating there was no statistical evidence of bias at the significance level 0.05. In other words, the hypothesis was accepted (hypothesis: the data was normally distributed), and hsa_circ_0004214 had normality characteristics. Accordingly, the 95% confidence interval (CI) for calculating the expression level of hsa_circ_0004214 was 15.64(LL)–18.81(UL) (values are log2 transformed). We considered the non-one-to-one relationship between the experimental group and the control group and took 95% CI ± 1 as the expression range of hsa_circ_0004214 in non-glioma normal brain tissue. The results showed that hsa_circ_0004214 was highly expressed in high-grade gliomas (WHO III/IV) compared with non-glioma controls and low-grade (WHO II) gliomas (Figure 3A, *p* < 0.05, *p* < 0.005). However, there was no statistical difference in hsa_circ_0004214 expression levels between WHO II glioma and non-glioma control group (Figure 3A, *p* > 0.05). Meanwhile, there was no difference in the expression level of hsa_circ_0004214 between WHO III and WHO IV (*p* > 0.05). The experimental results showed that the high expression of hsa_circ_0004214 may be correlated with the glioma high WHO grade. Compared with the non-glioma controls and low-grade (WHO II) gliomas, the expression of hsa_circ_0004214 was significantly higher in WHO III/IV gliomas.

### 3.4. Diagnostic Value of hsa_circ_0004214

For investigating the correlation of hsa_circ_0004214 levels with clinical parameters to evaluate a diagnostic potential, receiver-operating characteristic (ROC) curves were drawn using the SPSSAU system (https://spssau.com/index.html) (accessed on 11 September 2022). AUC is the area under the ROC curve. Its value ranges from 0 to 1. The closer the AUC is to 1, the better the diagnostic effect will be. The judgment standard of AUC is as follows: 0.5 or less does not meet the actual situation, 0.5 indicates no diagnostic value at all, 0.5–0.7 indicates low diagnostic value, 0.7–0.9 indicates certain diagnostic value, and above 0.9 indicates high diagnostic value. The AUC value of hsa_circ_0004214 expression was 0.880 (95% CI: 77.27%~98.73%), which indicated that hsa_circ_0004214 expression had certain diagnostic value for glioma (Figure 3B), and the corresponding optimal cut-off value was 0.667 (the sensitivity at this time was 0.767, the specificity was 0.900). A ROC curve was constructed for the expression of hsa_circ_0004214 to judge their diagnostic value for WHO III/IV (Table 3). The AUC value corresponding to hsa_circ_0004214 expression was 0.931 (95% CI: 85.49%~100.65%), which indicated that circ expression had high diagnostic value for WHO III/IV (Figure 3C), and the corresponding optimal cut-off value was 0.813 (the sensitivity at this time was 0.880, and the specificity was 0.933). Moreover, the difference of AUC value of the two clinical diagnostic effects was compared (independent standard). According to two groups of AUC values (AUC1 = 0.88, AUC2 = 0.931), two groups of standard errors (SE values, SE1 = 0.055, SE2 = 0.039), analyzed by the Hanley McNeil method. When *z*-values and *p*-values are combined to assess differences, *p* < 0.05 indicates a significant difference between the two ROC ranges (AUC). However, the results showed that there was no significant difference between the two ROC regions (*p* = 0.4494, Table 4).

Furthermore, the gliomas of different WHO grades were classified, and the frequency of diagnostic success and error were recorded, respectively (Figure 3D). hsa_circ_0004214 failed in 1 of 10 cases of normal brain tissue (0.1), 5 of 5 cases of WHO II (1), 3 of 10 cases of WHO III (0.3), and 0 of 15 cases of WHO IV (0). Diagnosis of low-grade glioma (including normal brain tissue controls) had 1 failure case in 15 cases (0.07). There were 3 negative cases among 25 cases (0.12) diagnosed as high-grade glioma (WHO III/IV).

### 3.5. Relationship of hsa_circ_0004214 Levels with Clinicopathological Factors of Patients with Glioma

The clinical data of 30 glioma patients were collected, including age, gender, height, body mass index (BMI), WHO grade, edema belt, position, pathological type and blood pressure. We analyzed whether the differential expression of hsa_circ_0004214 was related to the above factors (Table 5). The results showed that there was no significant relationship between the expression level of hsa_circ_0004214 and the maximum diameter of glioma, age, gender, height, BMI and tumor location but may be related to WHO grade (*p* < 0.05) and edema belt (*p* < 0.01). Further, the hsa_circ_0004214 expression is used as an independent variable, and the edema bandwidth is used as a dependent variable for linear regression analysis. Edema bandwidth = 1.669 + 0.001 hsa_circ_0004214 expression, and the R square value of the model was 0.001, which indicated that the circRNA expression could explain the 0.1% variation in edema bandwidth. When the F test was performed on the model, it was found that the model did not pass the F test (F = 0.023, *p* = 0.879 > 0.05), which means that the expression of hsa_circ_0004214 does not affect the bandwidth of edema. Therefore, it is impossible to specifically prove the relationship between the independent variable and the impact relationship. At the same time, we collected the immunohistochemical and genetic detection results of glioma patient samples, including Vimentin, S-100, GFAP, SYN, EMA, CD34, Ki67, IDH mutation, MEMG methylation and 1p/19q lost and other information (Table 6). The results indicated that the expression level of hsa_circ_0004214 had no significant relationship with Vimentin, S-100, SYN, EMA, CD34, IDH mutation, MEMG methylation and 1p/19q lost, but it might be related to Ki67 (*p* < 0.005) and GFAP (*p* < 0.005).

### 3.6. hsa_circ_0004214 Affects the Migration Capacity of Glioma In Vitro

In this study, we evaluated whether hsa_circ_0004214 is involved in the regulation of glioma cell behavior by changing the expression level of hsa_circ_0004214 and observing the changes in the migration ability of U87 cells. The hsa_circ_0004214 over-expression vector was designed, stably transfected into U87 cells, and the malignancy of tumor cells was further evaluated by analyzing the migration capacity of U87 cells (Figure 4A). After overexpressed with hsa_circ_0004214 (*p* < 0.05), the transwell assay indicated that the migration capacity of the transfected U87 cells was increased (Figure 4B). In addition, we constructed siRNA of hsa_circ_0004214 and down-regulated the expression of hsa_circ_0004214 by transfecting U87 cells. After knockdown of hsa_circ_0004214 in U87 cell (Figure 4C, *p* < 0.05), transwell assay showed that the migration capacity of U87 cells with low expression of hsa_circ_0004214 was decreased (Figure 4D). The results manifested that the number of U87 cells entering the transwell chamber was significantly increased in the OE-hsa_circ_0004214 group, compared with the control group vector. On the contrary, compared with the control group siRNA NC, the number of U87 cells entering the transwell chamber was significantly reduced in the siRNA-hsa_circ_0004214 group. We proved that the highly expressed hsa_circ_0004214 could promote the migration capacity of glioma. Based on the results of pathology, we speculated that the high expression of hsa_circ_0004214 may be related to the high mortality rate of patients. According to the results of transwell, it was considered that the differential expression of hsa_circ_0004214 may be related to the migration capacity of glioma cells.

### 3.7. Construction of Regulatory Network Associated with hsa_circ_0004214

MicroRNAs (miRNAs) are small non-coding RNA molecules, usually 21–25 nucleotides in length, that negatively regulate protein expression [11]. They are linked to cancer development and maintenance [12]. Some circRNAs specifically bind miRNAs as miRNA sponges to regulate gene expression, which can suppress the activity of miRNA and regulate the target genes [13]. This type of RNA is known as competitive endogenous RNA (ceRNA) [14]. We extracted calculated miRNA binding sites of conserved miRNAs and predicted the binding of hsa_circ_0004214 to miRNA and RBP. The miRNA microarray (GSE13030) Gene Expression Omnibus (GEO) dataset was included in this study, which studied the expression profiles of 340 miRNAs in 97 glioblastoma tissues, each hybridized with the respective adult non-neoplastic brain tissue as a control (Appendix A). Ten glioblastomas were randomly selected for analysis of miRNA expression. Up-regulated miRNAs are shown in red, and down-regulated miRNAs are shown in purple. The volcano plot indicates 33 down-regulated and 10 up-regulated miRNAs in glioblastoma, compared with the adult non-neoplastic brain tissue (Figure 5A). The heatmap visualizes the part of aberrantly expressed miRNA (Figure 5B). With the help of CSCD, we visualized the circRNAs and predicted the MRE and ORF. Firstly, we predicted a total of 237 potential target miRNAs of hsa_circ_0004214 from miRDB, ENCORI and circBANK. Further, we screened out the four most abnormally expressed miRNAs according to the miRNA microarray. At the same time, we predicted and screened the nine groups of RBP with the highest integrated score by the CSCD and ENCORI platforms. The information and possible mode of action of miRNA downstream target genes were obtained through NCDB and PubMed. Based on comprehensive data and some proven conclusions, the visualized interaction network of hsa_circ_0004214 was constructed by String and exhibited by Cytoscape 3.9.3 software (Figure 5C).

## 4. Discussion

The occurrence and development of glioma constitute a complex biological process. Molecular alterations of a large number of genes are involved in cancer progression to metastasis [15]. Although glioma has been extensively studied, the molecular mechanism of glioma occurrence and development has not been fully elucidated [16]. The expression of non-coding RNAs, such as miRNAs and lncRNAs, have been applied and considered as feasible and useful methods for analyzing the molecular characteristics of glioma [17]. Today, regulatory networks of circRNAs have expanded our understanding of the complexity of noncoding RNAs [18]. In this study, for the first time, hsa_circ_0004214 was found to be significantly upregulated in glioma, and proposed as a potential and stable biomarker to diagnose this particular cancer type. Altogether, these findings support that circRNAs play important roles in cancer, and they may serve as new strategies for diagnosis and treatment. Although the current understanding of circRNAs expression is still in its infancy, this work highlights its potential and provides guidance for the development of future methods for clinical evaluation.

One of the most extensive studies is circRNAs terminating the regulation of miRNA to its target gene by binding to the miRNA as a competing endogenous RNA (ceRNA) through the base complementary pairing principle [19]. According to other reports, expression of the hsa_circ_0004214 is typically up-regulated in OLP-Associated Oral Squamous Cell Carcinoma (OSCC) [16] and Cervical Cancer (CC) [20] and may act as a molecular sponge for miRNAs to influence gene expression in tumors. This study is the first to verify the expression level of hsa_circ_0004214 in glioma. To obtain the full nucleic acid sequence of hsa_circ_0004214 through sequencing primers were designed according to the splicing characteristics of circular RNA, confirming that hsa_circ_0004214 forms a circle at the splicing level. hsa_circ_0004214 was marked with FISH probe, and the results showed that hsa_circ_0004214 was mainly distributed in the cytoplasm. The results indicated that the expression level of hsa_circ_0004214 in the glioma core tissue was significantly higher than that in the paracancerous tissue. In addition, we found that the levels of hsa_circ_0004214 expression in glioma tissues were increased compared with that in non-tumor brain tissues. There was no significant difference in the expression level of hsa_circ_0004214 between the paracancerous tissue and the non-tumor brain tissue control group. Experimental results suggested that hsa_circ_0004214 can be stably overexpressed in glioma. However, whether up-regulated expression of hsa_circ_0004214 in glioma also plays the same function as in OSCC and CC remains to be further explored.

Studies have reported that circRNAs can be used as a diagnostic marker for tumors. For example, circNHSL1 was confirmed to be a highly stable circRNA and an appropriate diagnostic and prognostic marker for gastric cancer [21]. Compared with serum CEA level, hsa_circ_0000745 has higher sensitivity and specificity in gastric cancer screening. However, the combination of hsa_circ_0000745 plasma with CEA serum improves the diagnostic value [22]. According to the 2021 edition of WHO’s guidelines for grading gliomas, gliomas are divided into 4 grades, and the common ones for adults are WHO II-IV. The expression levels of hsa_circ_0004214 in different grades of glioma were analyzed. The results revealed that there was significant differential expression between low-grade glioma (WHO II) and high-grade glioma (WHO III/IV). We found that expression levels of hsa_circ_0004214 in glioma tissues were associated with tumor WHO grade. hsa_circ_0004214 can be used as a molecular target to assess the prognosis of glioma, helping to assess the degree of tumor malignancy. In addition, we also determined its potential diagnostic value and reported here for the first time that hsa_circ_0004214 has high sensitivity and specificity in the screening of glioma. It must be pointed out that hsa_circ_0004214 performed poorly in diagnosing WHO II glioma. However, it has high diagnostic value in the application of high-grade glioma. These results suggested that hsa_circ_0004214 may be associated with genetic alterations in different grades of glioma. In the study of the correlation between hsa_circ_0004214 and many pathological factors, only a positive correlation was found with the width of the tumor edema band. This may be related to the degree of malignancy of the tumor, and the specific correlation with hsa_circ_0004214 is unknown. In the correlation analysis of immunohistochemical experiments and genetic testing results, hsa_circ_0004214 was only correlated with GFAP (+) and Ki67 (+), but Vimentin, S-100, SYN, EMA, CD34, IDH mutation, MEMG methylation and 1p/19q loss was not significantly correlated. It should be emphasized that the correlation between hsa_circ_0004214 and GFAP may be interfered by the source of glioma tissue. Some statistics show that the positive rate of GFAP in the diagnosis of glioma is approximately 85–95% [23]. GFAP negativity depends mainly on glioma histological origin or other factors [24,25]. For example, GFAP may be negative in gliomas of non-astrocyte origin but positive in high-grade gliomas with admixture of differentiated astrocytic cells. In addition, some GFAP-negative samples in this experiment were oligodendroglioma (WHO II), and circ0004214 was also lowly expressed in these samples. This may also affect the reliability of the conclusion that circ0004214 is related to GFAP. In this study, hsa_circ_0004214 is used to assist immunohistochemical diagnosis of glioma grading, but its specific quantitative relationship with GFAP and Ki67 needs to be further explored in the future, and the underlying mechanisms of how hsa_circ_0004214 regulates Ki67 and GFAP expression and cell proliferation were unclear.

To assess whether hsa_circ_0004214 is involved in the regulation of glioma cell behavior. The hsa_circ_0004214 over-expression vector and siRNA were constructed and transfected into U87, respectively. Transwell showed that compared with the control group, U87 overexpressing hsa_circ_0004214 had a stronger migration capacity. On the contrary, the migration capacity of U87 with reduced expression of hsa_circ_0004214 was suppressed. It is suggested that hsa_circ_0004214 can affect the migration capacity of glioma and has the potential as a prognostic factor. According to the results of transwell, the high expression of hsa_circ_0004214 in high-grade glioma reflects its high activity in the migration process of high-grade glioma. It is well known that miRNAs can inhibit translation or reduce mRNA stability by directly targeting gene 3′-UTR, thereby acting as a post-transcriptional regulator. We predicted a ceRNA network based on hsa_circ_0004214 covering 4 miRNAs and 9 RBPs. We further established a prognosis- associated ceRNA subnet consisting of 1 circRNAs, 4 miRNAs, 9 RBPs and 11 mRNAs. The relationship between all miRNAs and downstream target genes has been experimentally determined. The mechanism of miRNA and RBP interaction with hsa_circ_0004214 remains to be further verified. In summary, our study revealed that intraoperative detection of circular RNA hsa_circ_0004214 may serve as a diagnostic marker for high-grade glioma, supplementing to the immunohistochemical diagnosis of glioma.

This study comprehensively discussed the expression of hsa_circ_0004214 between the control group and different groups of glioma, and counted the expression of the adjacent tissues of glioma and the non-glioma control group of non-glioma patients. The possibility of hsa_circ_0004214 as a glioma marker was evaluated and the diagnostic success and failure ratios were calculated according to different perspectives. Regretfully, according to the results of this study, it can only be found that hsa_circ_0004214 is abnormally expressed in high-grade glioma tissue. Although this can be used as auxiliary evidence for immunohistochemistry, surgery is still required to remove the tumor tissue. Accurately diagnosing glioma without surgery has always been our pursuit, which at least requires markers that can be secreted in cerebrospinal fluid [26]. Even better, special markers can be detected in blood [27], which requires it to have the ability to cross the blood-brain barrier [28]. Therefore, whether hsa_circ_0004214 can be differentially expressed in cerebrospinal fluid as exosomes is the direction that we need to continue to explore in the future. Judging the WHO grading of glioma before surgery can guide the selection of surgical methods and can predict and prepare in advance for possible unexpected situations during surgery. CircRNAs are often widely studied as markers of tumor prognosis [29,30]. We cannot deny that circular RNAs have powerful functions in modulating tumor, they are not widely studied as tumor diagnostic markers. However, unlike other tumors, glioma rarely metastasizes, and its prognosis is closely related to WHO grading [31,32,33]. Finding an excellent tumor marker is a must. CircRNAs are different from other linear RNAs because their circular structure can resist the degradation of exonuclease and have a longer half-life; in principle it is a heaven-born marker [34]. Of course, hsa_circ_0004214 is not necessarily the best circRNA marker, but we want to verify that circRNAs can be used to assist in the diagnosis of glioma through the example of hsa_circ_0004214. It is believed that in the future, in addition to the tumor regulation field of circRNAs, its diagnostic value in special tumors can be more widely concerned.

## 5. Conclusions

In our study, we found that hsa_circ_0004214 has a high diagnostic value in judging high-grade glioma, and hsa_circ_0004214 can promote the migration process of glioma cells in vitro. At the same time, we successfully constructed a hsa_circ_0004214 centered circ RNA regulatory network.

## Figures and Tables

**Figure 1 brainsci-13-00193-f001:**
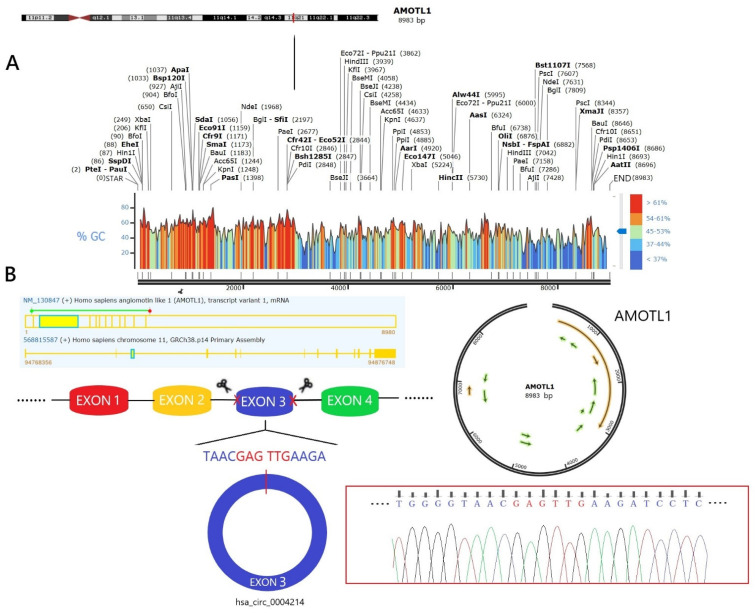
Genetic information of host gene AMOTL1 and hsa_circ_0004214. (**A**) Genetic information of host gene AMOTL1. Full-length gene visualization analysis, including base pair annotation, the location of host gene in Genome-wide, GC ratio (%) and common commercial targets. (**B**) Genetic information of hsa_circ_0004214. Containing host gene exon positions, circRNA alternative splicing, circular docking sequence and basic biological information.

**Figure 2 brainsci-13-00193-f002:**
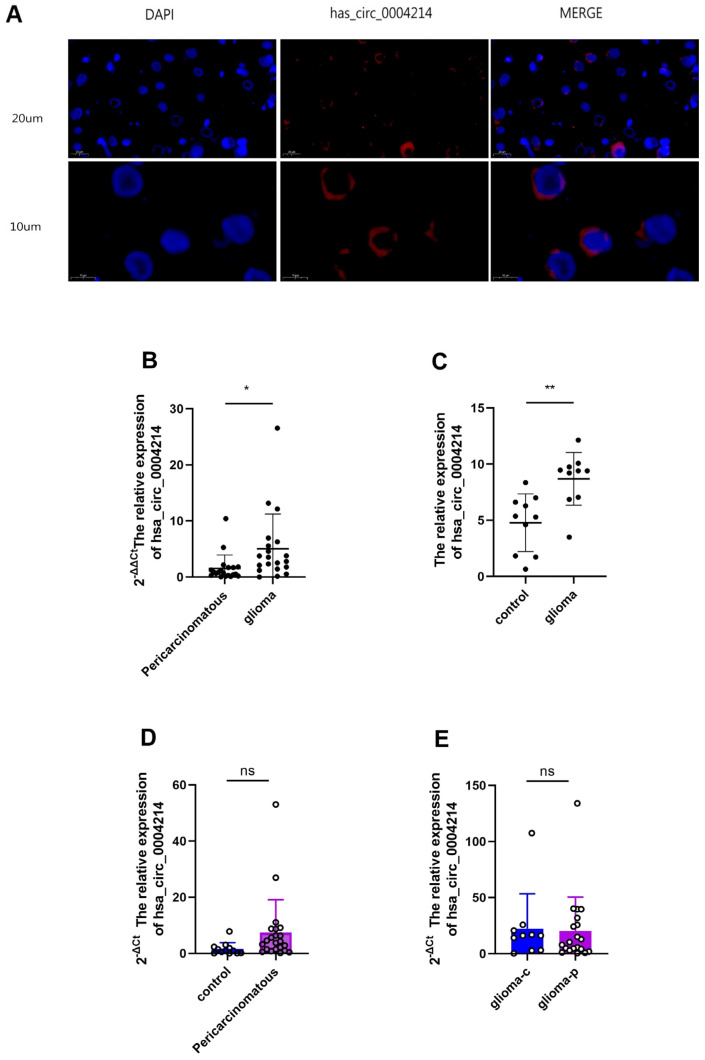
Subcellular localization of hsa_circ_0004214 in tumor cells and expression level in tissues. (**A**) RNA FISH detecting hsa_circ_0004214′s subcellular localization in U87 cells. Nuclei was stained with DAPI. Scale bar, 20 μm and 10 μm. (**B**) Expression level of hsa_circ_0004214 evaluated by RT-qPCR in paratumor tissue and glioma core tissue (*n* = 20, no significant (ns): *p* > 0.05, * *p* < 0.05, ** *p* < 0.01). (**C**) Expression of hsa_circ_0004214 evaluated by RT-qPCR in non-glioma control group and glioma group (*n* = 10). (**D**) Expression of hsa_circ_0004214 evaluated by RT-qPCR in paracancerous tissues and non-glioma control group (*n* = 10, *n* = 10). (**E**) Expression of hsa_circ_0004214 evaluated by RT-qPCR between two groups glioma core tissue (*n* = 20, *n* = 10).

**Figure 3 brainsci-13-00193-f003:**
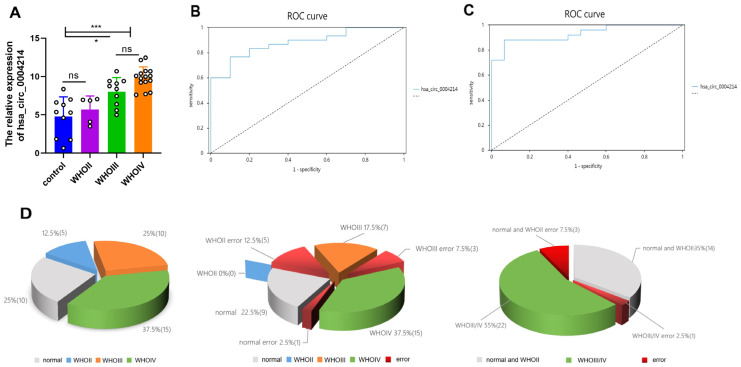
Expression level and diagnostic value of hsa_circ_0004214 in different grades of glioma. (**A**) Expression level of hsa_circ_0004214 in different grades of glioma (* *p* < 0.05, *** *p* < 0.005). (**B**) hsa_circ_0004214 distinguishes non-glioma brain tissue from control group and glioma tissue, and its ROC curve. (**C**) hsa_circ_0004214 distinguishes low-grade glioma (control + WHO II) and high-grade glioma (WHO III/IV), and its ROC curve. (**D**) Evaluation of data for different diagnostic categories via pie charts. The frequency of diagnostic success and error were recorded, respectively.

**Figure 4 brainsci-13-00193-f004:**
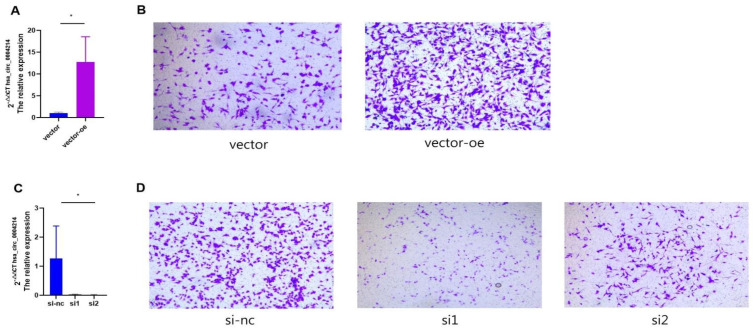
Judging the change of glioma cell migration capacity by changing the expression level of hsa_circ_0004214 in U87. (**A**) U87 cells transfected with hsa_circ_0004214 overexpression plasmid for 48 h, and the level of hsa_circ_0004214 was detected by RT-qPCR (* *p* < 0.05). (**B**) Transwell assay detected the migratory capacities of U87 cells after up-regulated the level of hsa_circ_0004214. Scale bar, 100 μm (* *p* < 0.05). (**C**) U87 cells transfected with two siRNAs specifically targeting hsa_circ_0004214(si1, si2) for 48 h, and the level of hsa_circ_0004214 was detected by RT-qPCR. (**D**) Transwell assay detected the migratory capacities of U87 cells after down-regulated the level of hsa_circ_0004214. Scale bar, 100 μm.

**Figure 5 brainsci-13-00193-f005:**
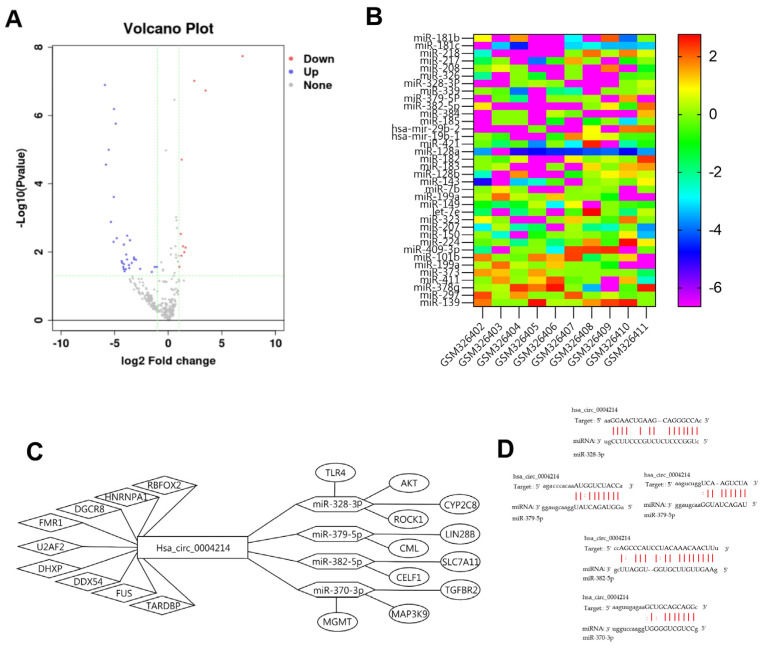
Construction of regulatory network associated with hsa_circ_0004214. (A) The volcano plot showing the expression profile of miRNAs in glioma tissue samples. (**B**) The heat map showing the top 20 down-regulated miRNAs and 10 up-regulated in glioma tissue samples compared with non-glioma brain tissue samples analyzed by RNA sequencing. (**C**) Construction of a RBP/circRNA/miRNA/mRNA ceRNA Interaction network. The network consists of hsa_circ_0004214, 4 miRNAs, 12 RBPs, and 11 mRNAs. (**D**) From ENCORI, we predicted binding sites of miR-28-3p, miR-379-5p, miR-382-5p, and miR-370-3p with hsa_circ_0004214.

**Table 1 brainsci-13-00193-t001:** Target information.

Target ID	Type	Direction	Sequence (5′→3′)	Product Length
hsa_circ_0004214	①Divergent	Forward primer	GTGAGAACAGATGTGGCCGT	138
		Reverse primer	GCATCCTCGGAACCTCTCATT	
	②Convergent	Forward primer	AGCCCATCCTACAAACAA	218
GAPDH		Reverse primer	CCTGAAGAACTGCGACTG	
		Forward primer	CTCGCTTCGGCAGCACA	122
		Reverse primer	AACGCTTCACGAATTTGCGT	

**Table 2 brainsci-13-00193-t002:** Primer.

Target ID	Official Symbol	Primary Source	Gene Type	Organism	Location
Angiomotin like 1	AMOTL1	Nucleotide: NM_130847	protein coding	Homo sapiens	11q21
hsa_circ_0004214	CircAMOTL1	GEO: GSE146463	circ-RNA	hsa	chr11:94532555-94533477(+)

**Table 3 brainsci-13-00193-t003:** Diagnostic value of hsa_circ_0004214.

Group	*n*	Mean + SD(∆∆CT)	Specificity	Sensitivity	Youden Index	OR	RR
Pericarcinomatous and glioam tissue	20	−1.24 ± 2.39	0.8	0.8	0.6	16	4
Control and glioma	10	−1.33 ± 2.35	0.9	0.7	0.6	21	7
Contor-WHO II and WHO III/IV	40	−1.75 ± 1.81	0.933	0.88	0.813	120.667	13.2

**Table 4 brainsci-13-00193-t004:** ROC results AUC summary.

The Title	Goal	AUC	Standard Error (SE)	*p*	95% CI	Best Threshold	Sensitivity	Specificity	Cut-Off	AUC Difference	Z	*p*
hsa_circ_0004214 level	glioma	0.880	0.055	0 < 0.05	0.773~0.987	0.667	0.767	0.767	−14.990	0.0510	0.7564	0.4494
hsa_circ_0004214 level	WHOIII/IV	0.931	0.039	0 < 0.05	0.855~1.006	0.813	0.880	0.933	−14.950			

**Table 5 brainsci-13-00193-t005:** Relationship of hsa_circ_0004214 expression levels (ΔCt) in glioma tissues with clinicopathological factors.

Clinicopathological Factor		*n*	Positive	Negative	Tissue hsa_circ_0004214	
					Mean ± SD	*p* Value
Age (yr)	<60	16	11	5	13.13 *±* 1.95	0.5135
	≥60	14	11	3	13.67 *±* 2.48	
Sex	Male	16	14	2	12.75 ± 1.57	0.0795
	Female	14	8	6	14.18 ± 2.65	
Male height (cm)	<170	5	5	0	13.04 ± 1.30	0.2930
	≥170	11	9	2	12.12 ± 2.07	
Female height (cm)	<160	4	2	2	14.13 ± 2.72	0.9112
	≥160	10	6	4	14.31 ± 2.86	
BMI	18.5–23.9	22	17	5	13.12 ± 2.45	0.2307
	>23.9	8	5	3	14.24 ± 1.22	
	<18.5	0	0	0	-	
WHO grade	II	5	0	5	16.31 ± 1.74	0.0382(II:III)
	III	10	7	3	13.99 ± 1.85	0.0001(II:IV)
	IV	15	15	0	12.17 ± 1.42	0.0281(III:IV)
Diameter in (cm)	<4	12	7	5	13.99 ± 2.75	0.2552
	≥4	18	15	3	13.04 ± 1.78	
edema belt (cm)	<1	14	6	8	14.74 ± 2.23	0.0014
	≥1	16	16	0	12.34 ± 1.46	
Position	Frontal lobe	12	7	5	13.99 ± 2.42	0.8016(F:T)
	Temporal lobe	12	9	3	13.42 ± 2.35	0.3658(F:O)
	Other position	6	6	0	12.48 ± 1.05	0.6700(T:O)
pathological type	astrocytic glioma	14	8	6	14.24 ± 2.20	0.7182(A:O)
	glioblastoma	13	13	0	12.22 ± 2.19	0.0308(G:A)
	Other type	3	1	2	14.97 ± 2.65	0.0595(G:O)
Blood Pressure (mmHg)	<140	16	13	3	12.79 ± 2.12	0.0721
	≥140	14	9	5	14.23 ± 2.08	

**Table 6 brainsci-13-00193-t006:** Relationship of hsa_circ_0004214 levels with Genetics factors of patients with glioma.

Items		*n*	Positive	Negative	Tissue hsa_circ_0004214	
					Mean ± SD	*p* Value
immunohistochemical	Vimentin (+)	28	20	8	13.48 ± 2.22	-
	Vimentin (−)	0	0	0	-	
	s-100 (+)	23	19	4	13.05 ± 1.66	0.0586
	s-100 (−)	7	3	4	14.83 ± 3.21	
	GFAP (+)	24	22	2	12.86 ± 1.75	0.0002
	GFAP (−)	6	0	6	16.23 ± 1.66	
	SYN (+)	25	19	6	13.25 ± 2.28	0.1283
	SYN (−)	5	3	2	14.90 ± 0.93	
	EMA (+)	14	13	1	12.64 ± 1.78	0.2962
	EMA (−)	11	8	3	13.45 ± 1.96	
	CD34 (+)	17	14	3	12.84 ± 1.81	0.9391
	CD34 (−)	9	7	2	13.51 ± 2.02	
	Ki67 (+)	22	19	3	12.71 ± 1.77	0.0008
	Ki67 (−)	8	3	5	15.53 ± 1.96	
Genetics	IDH mutation (+)	11	6	5	14.42 ± 1.86	0.0779
	IDH mutation (−)	19	16	3	12.97 ± 2.22	
	MEMG methylation (+)	6	5	1	13.92 ± 1.65	0.6040
	MEMG methylation (−)	24	17	7	13.40 ± 2.31	
	1p/19q lose (+)	0	0	0	-	-
	1p/19q lose (−)	30	22	8	13.46 ± 2.19	

## Data Availability

Upload before publishing.

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
