# Peer review of "A Meaningful Strategy for Glioma Diagnosis via Independent Determination of hsa_circ_0004214"

_brainsci, 2023, doi:10.3390/brainsci13020193_

Round 1

Reviewer 1 Report

he authors elucidated expression levels of the circular RNA, hsa_circ_0004214 in glioma tissues and non-tumor control or paraneoplastic tissues and demonstrated that WHO grade II glioma showed significantly higher expression of the circular RNA than non-tumor control. Although WHO grade III/IV much more highly expressed hsa_circ_0004214 than low grade glioma (grade II), there was no significant difference in the expression of the circular RNA between grade III and IV. Consequently, measurement of hsa_circ_0004214 in the serum and in the CSF may not be useful for diagnosing a high grade of gliomas and predicting the prognosis of patients with high grade gliomas. In addition, the authors disclosed that hsa_circ_0004214 had an enhancing ability in glioma migration.

The manuscript is well written, and the contents are highlighted roles of circular RNA, hsa_circ_0004214 in gliomas. But several points in the manuscript should be required correction as follows.

#1 In the analysis of the mRNA expression of hsa_circ_0004214, authors examined expression of tumor core tissues and paratumoral tissues. Why did the authors select the tumor core? Also, the authors obtained the tissues of paraneoplastic areas (at least 1cm away from the tumor), how did the authors accurately determine the target regions? Navigation-guided surgery is applied to resect the tumors? 

#2 The authors described there was significant difference in the expression of hsa_circ_0004214 between grade III and IV. These results might show hsa_circ_0004214 does not participate in an increase of malignancy in gliomas. Positive expression of hsa_circ_0004214 in low grade glioma may support the low activity of tumor progression in the hsa_circ_0004214-regulated system. Possible relation between high expression of hsa_circ_0004214 and glioma malignancy should be discussed.  

#3 Expression level of hsa_circ_0004214 had significant relation with expressions of GFAP and Ki67. Aside of Ki67, high expression of GFAP represents well-differentiated gliomas. This may be a reason there was no difference in the hsa_circ_0004214 expression between grade III and grade IX gliomas. The authors should describe histological features of gliomas showing a high expression of GFAP.

#4 Abstract, line 20. the authors described “the invasion ability of U87 was enhanced after “

Page 11, line 312, “hsa_circ_0004214 may be related to the invasion ability of glioma cells”, but invasion study was not included in the present study. To analyze in vitro invasion of tumor cells, extracellular matrix components such as Matrigel is necessary for the transwell invasion assay. 

#5 Do the authors have an idea concerning molecular mechanisms why hsa_circ_0004214 can enhance glioma migration and maybe glioma invasion.

#6 There are several mistake in sentences such as lines 46-47in Introduction (no verb), lines 370-371 (no verb), and others. 

Reviewer 2 Report

The manuscript examines the role of the circular RNA hsa_circ_0004214 as a potential diagnostic tool in malignant glioma. It presents novel data based on reliable methodology and results.

However, there are some minor issues to be clarified basically in the Materials and Methods section. Cell cultures should precede the Transwell migration assay, as coming first it is not clear which cells are meant, are they transfected or not, etc. Additionally, the transfection procedure is not described – there is just a sentence (lines 125-126) which is incomplete and unclear. The text from the legend of Figure 4 could come here in some more detail.

A basic achievement of the authors is the constructed regulatory network associated with hsa_circ_0004214. The bioinformatics approach which is adequate to the aim of the study and performed professionally, revealed 237 potential target miRNAs. Finally, only 4 of them and 9RBPs were found to be most likely to interact with the circRNA of interest.

The Discussion is well focused and based on original data. It could be more convincing if the limitations of the study are also stated.

The English language style should be edited as there are several incomplete sentences, missing verbs or incorrect grammar usage (lines 46-47; 125-126; 228; 441; etc.) or typing or spelling errors (209; 309; etc.).

As a whole, the manuscript presents new and interesting results.
